# Bupivacaine as a euthanasia agent for African Clawed Frogs (*Xenopus laevis*)

**Kaela Navarro**[1]*, **Katechan Jampachaisri**[2‡], **David Chu**[1‡], **Cholawat Pacharinsak**[1]

**1** Department of Comparative Medicine, Stanford University, Stanford, California, United States of America,
**2** Department of Mathematics, Naresuan University, Phitsanulok, Phitsanulok Province, Thailand

☯ These authors contributed equally to this work.
‡ KJ and DC also contributed equally to this work.
* kaela.cab92@gmail.com

## Abstract

Immersion in tricaine methanesulfonate (i.e. TMS) has been used for euthanasia of *Xenopus laevis* (African Clawed frogs). However, the time for preparation and potential human health hazards may pose as a barrier for large group culls. Here, we aimed to investigate whether immersion in bupivacaine is an effective means to euthanize this species. In experiment one, frogs (n = 10/group) were randomly assigned to 1-h immersion in 1 of 3 treatment groups: 1) *TMS-5* (MS-222, 5g/L); 2) *TMS-10* (MS-222, 10 g/L); or 3) *Bupi-1.5* (0.5% Bupivacaine, 1.5 g/L). Frogs were then removed from solutions, rinsed with system water, and placed into a recovery cage. Heart rate was evaluated audibly via doppler ultrasound flow over 1 min at immediate removal (T1h), at 2 (T2h), and 3 (T3h) h in the recovery cage. In experiment two, frogs (n = 7/group) underwent 5-h & 19-h immersion in either *TMS-5* or *Bupi-1.5*, with heart rate assessment at 5 and 19 hrs. Righting reflex and withdrawal reflex of the hindlimb were tested during the experiments. *Experiment one*—after the 1-h immersion, *Bupi-1.5* treated animals had decreased heart rates compared to *TMS-5* and *TMS-10* treated animals by T2h. Neither *TMS-5*, *TMS-10*, nor *Bupi-1.5* ceased heart rate after the 1-h immersion. *Experiment two*—after the 5-h immersion, *Bupi-1.5* and *TMS-5* treated animals were comparable in heart rates. 43% of *TMS-5* animals and 14% of the *Bupi-1.5* animals had completely ceased heart rates at T5h. At 19 h all remaining animals exhibited rigor mortis and had ceased heart rate. We recommend 19-h of immersion using either TMS-5 or Bupi-1.5 for cessation of heart rate in African Clawed frogs. These data are strong support for the use of secondary physical methods for euthanasia in African Clawed frogs when euthanasia by immersion is performed.

## Introduction

Euthanasia is a necessary task in the field of laboratory animal medicine and animal research. The vast majority of animals utilized in research are mice and rats, thus much is known about welfare, physiologic processes, and techniques employed for euthanasia of these species [1]. However, non-mammalian species are also frequently involved in research and *Xenopus laevis*

**Data Availability Statement:** All data has been included within the paper text and its Supporting Information files.

**Funding:** This work was supported by Stanford's Department of Comparative Medicine. The funders

had no role in study design, data collection and analysis, decision to publish, or preparation of the manuscript. The content is solely the responsibility of the authors and does not necessarily represent the official views of the Stanford Department of Comparative Medicine or any other agency of the State of California.

**Competing interests:** The authors have declared that no competing interests exist.

(African Clawed frogs) are one such species. African Clawed frogs are animal models used experimentally in the fields of developmental biology, endocrinology, and toxicology [2]. They are also an attractive model for their relative ease of husbandry, inducible egg production, and short generation time. These animals are commonly housed in colonies and at times culls of large groups of animals may be necessary, such as in mass disease outbreak or in the emergency of an aquatic system failure. The process of euthanasia for these frogs can be challenging to perform due to differences from rodents, such as anatomical landmarks used for certain euthanasia methods, lack of grimace scales to determine stress or pain, difference of metabolism of drugs, and the ability to hold their breath underwater.

Despite these differences, an effective and humane euthanasia is still warranted. The 2020 AVMA Euthanasia Guidelines advocate for euthanasia techniques incorporating rapid loss of consciousness, followed by cardiac or respiratory arrest, and ultimately loss of brain function, while minimizing distress prior to loss of consciousness [3]. Current recommendations of primary euthanasia techniques for amphibians include intravenous or intracoelomic injection of sodium pentobarbital or physical penetrative methods, such as decapitation followed by pithing, after being anesthetized [3, 4]. Euthanasia by sodium pentobarbital injection not only requires high dosages, and shows inconsistent time until loss of consciousness is achieved, but can be laborious to perform in a reasonable time period for larger groups of animals, such as might be warranted in a disease outbreak scenario or aquatic system failure scenario [5]. The physical penetrative methods must be performed under anesthesia, can be technically challenging, and can contribute towards compassion fatigue in animal care personnel [2, 3]. Immersion in a drug containing solution is one route which can be utilized to circumvent the challenges faced with the previously described methods [5]. As aquatic amphibians have semi-permeable skin that allows for water soluble drugs to be metabolized through absorption, immersion in a solution of a euthanasia agent provides a potential euthanasia method [6–8].

Tricaine methanesulfonate (MS-222, TMS) is a derivative of the local anesthetic benzocaine, and has sodium-channel blocking activity, preventing neural transmission of mechanosensory stimuli. Immersion in a solution of TMS induces general anesthesia in aquatic animals, the dose of which is dependent on species and life stage [8–11]. It has also been investigated and used for euthanasia purposes in fish and amphibians. Immersion in TMS at 5 g/L for at least 1 h is described within the American Veterinary Medical Association (AVMA) Euthanasia Guidelines as a method for euthanasia of African Clawed frogs [3, 5]. However, use of TMS has drawbacks: it has a pH of 3.0, which can be irritating to the animal's skin, so it must be buffered to neutral pH before use [12]. Additionally, TMS exposure for animal care personnel has been associated with retinal toxicity and irritation to the respiratory tract if either the powder or vapor fumes of the liquid are inhaled posing a human health hazard [13, 14]. Thus, using an alternative drug mixed into water for euthanasia purposes may provide the same advantages as TMS without the labor or human health hazard challenges.

In this study we compared the effect of bupivacaine, a local anesthetic, to TMS on heart rate during the euthanasia process to assess its potential as a euthanasia agent. We chose bupivacaine because it is a non-controlled substance, does not require extensive preparation like TMS, and does not pose the same exposure toxicity as TMS to humans. *In vitro* experiments assessing the effect of bupivacaine on atrial fibers of frogs have shown the drug to have a negative chronotropic effect [15, 16]. We also observed this effect on heart rates of African Clawed frogs in a pilot study. Thus, we hypothesized immersing African Clawed frogs in bupivacaine at a concentration of 1.5 g/L would have an equal effect on heart rate as TMS at 5 g/L.

## Materials and methods

### Animals and husbandry

Adult female African clawed frogs (*Xenopus laevis*, n = 44; Nasco, Fort Atkinson, WI) were utilized. The animals used in this study were non-experimental retired egg layers. Animals were group housed in recirculating water tanks at a density of 1 frog per 4 liters water with a PVC pipe for enrichment and fed a commercial diet (Frog Brittle, Nasco) twice weekly at approximately one gram per frog. Daily water quality assessments were made and quality was maintained within the following parameters: water temperature between 16–21˚C, pH between 6.5–8.5, conductivity between 500–3000 μS, ammonia ($NH_3$) less than 0.05mg/L, nitrite ($NO_2^-$) < 0.1 mg/L, and nitrate ($NO_3^-$) < 50 mg/L. The frog colony was PCR negative for *Batrachochytrium dendrobatitidis*, *Mycobacterium chelonae*, *M. gordonae*, *M. marinum*, *Pseudocapilaroides xenopi*, and *Ranavirus*. All experiments were approved by Stanford University's Administrative Panel for Laboratory Animal Care. All frogs were raised in accordance with the *Guide for the Care and Use of Laboratory Animals* in a facility accredited by the Association for the Assessment and Accreditation of Laboratory Animal Care, International.

### Study design

We consulted with a statistician (KJ) during experimental design to determine the number of animals needed to generate statistical significance. All animals were in appropriate physical condition and health as deemed by institutional veterinarians, thus none were excluded from study. For experiment 1, evaluators were blinded to all experimental groups; groups were revealed at the end of the experiment. Due to the number of groups in experiment 2, euthanasia solutions were known. All solutions had a total volume of 0.5 L/frog in a transparent plastic container. The pH of each solution was measured using a handheld pH probe (Hanna Instruments, Woonsocket, RI) and buffered to a neutral pH using sodium bicarbonate (Doctors Foster & Smith, Rhinelander, WI), as needed. The final pH of all solutions ranged from 7–7.05. Animals were immersed in solution at staggard time points to allow for adequate heart rate & reflex assessments. Tank location was not accounted for in possible confounders. *Experiment one– 1 hour immersion)* To assess the effect of 1-h immersion on heart rate, frogs from three separate tanks were transferred to a holding bin and then chosen at random for 1 of 3 euthanasia solutions (30 frogs total; n = 10/group): 1) *TMS-5* (5g/L MS-222, Syndel, Ferndale, WA); 2) *TMS-10* (10 g/L MS-222); or 3) *Bupi-1.5* (1.5 g/L bupivacaine, Hospira Inc., Lake Forest, IL). At the end of 1-h immersion, animals were removed from the euthanasia solutions, rinsed with fresh source water, replaced into a rinsed & emptied container, and kept moist with moistened paper towels. Heart rates (bpm) were evaluated audibly via doppler ultrasound flow at 1 h (T1h, the end of 1 h immersion), 2 (T2h), and 3 (T3h) h. Time for this assessment ranged from 20–30 min. At the end of the experiment, the animals were placed in dorsal recumbency for the last heart rate assessment. The withdrawal reflex was re-assessed & a celiotomy was then performed to visualize the heart; the animals were then euthanized via cardiectomy (Fig 1A). No animals were excluded from experiment one. *Experiment two– 5 and 19 hour immersion)* To assess the effect of an extended immersion time on heart rate, frogs from two separate tanks were randomly assigned to 1 of 2 euthanasia solutions (14 frogs total; n = 7/group): 1) *TMS-5* (5g/L MS-222) or 2) *Bupi-1.5* (1.5 g/L bupivacaine). Frogs were immersed in either of these solutions for 5 h (T5h). Frogs were removed from solutions at T5h to perform heart rate assessment with the same method as experiment 1. Time for this assessment ranged from 15–20 min. Frogs with heart beats still present after the T5h evaluation were replaced into their respective euthanasia solutions for a total immersion time of 19 h. A celiotomy was performed

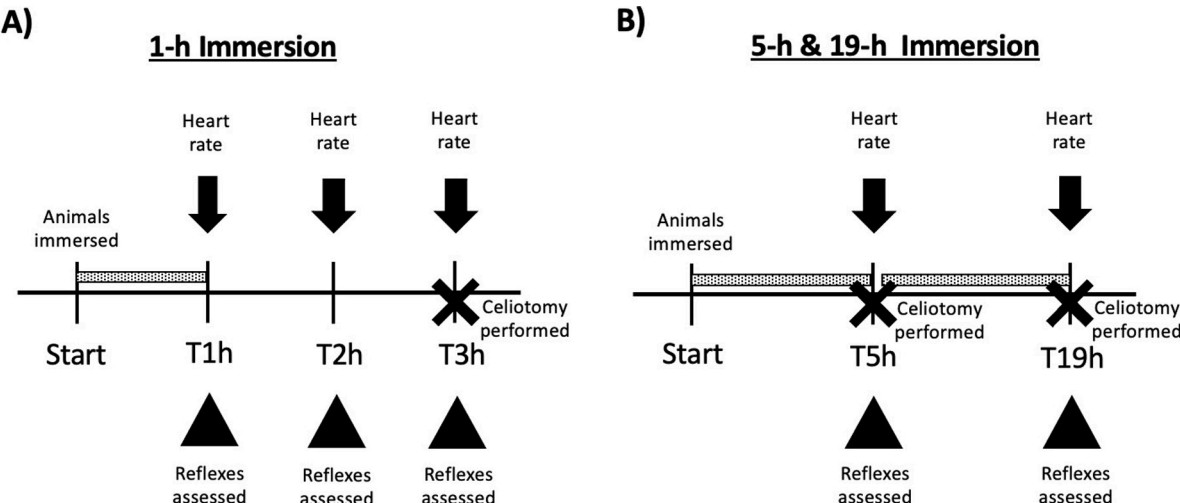

**Fig 1. Study design schematic.** A) 1-h immersion. B) 5-h & 19-h immersion. Dotted bar indicates length of time animals were immersed in solution. Heart rate assessment was performed via doppler ultrasound flow with the probe placed over the heart. Reflex assessment included withdrawal reflex of the right hindlimb and righting reflex. For A) a celiotomy was performed for all animals at the end of the experimental treatment. For B) a celiotomy was performed for frogs without audible heart beats using doppler ultrasound flow at 5-h of immersion & the remaining animals at 19-h of immersion.

on frogs with ceased heart rates to confirm visual lack of heart beat and a cardiectomy was performed. Remaining frogs were removed from euthanasia solutions the next morning and evaluated visually for rigor mortis and for heart rate via celiotomy (Fig 1B).

### Reflex assessment

Withdrawal reflex was performed once for both experiment one and two at 1 h and 2.5 h, respectively. To test withdrawal reflex to mechanical stimulus, the interdigital webbing of the right hindlimb was pinched once with mosquito hemostats. Righting reflex was performed each time heart rate assessment was performed for both experiments. To test righting reflex, frogs were turned to dorsal recumbency, if not already in this position, and observed for the ability to turn to ventral recumbency.

## Results

### 1 hour immersion

There was no significant difference between any treatment group in heart rates after 1 h of immersion. The heart rate of animals immersed in Bupi-1.5 were significantly ($P < 0.05$) decreased compared to TMS-5 & TMS-10 g/L at 2h ($10.8 \pm 0.9$, $17.4 \pm 0.8$, $17.4 \pm 2.6$ bpm, respectively). At 3 h post-immersion Bupi-1.5 was significantly decreased to only TMS-5 ($8.2 \pm 1.0$ and $16.6 \pm 1.1$, respectively). There was no significant difference between the heart rates of animals in TMS-5 or TMS-10 at any time point (Fig 2).

### 5 and 19 hour immersion

There was no significance difference between the TMS-5 and Bupi-1.5 groups at 5 h of immersion. At 5 h of immersion, we could not auscultate heart sounds in 3 of the TMS-5 treated frogs (43%) and 1 of the Bupi-1.5 treated frogs (14%); gross necropsy was performed, and the animals had zero visible cardiac contractions (Table 1). All animals with an audible heartbeat

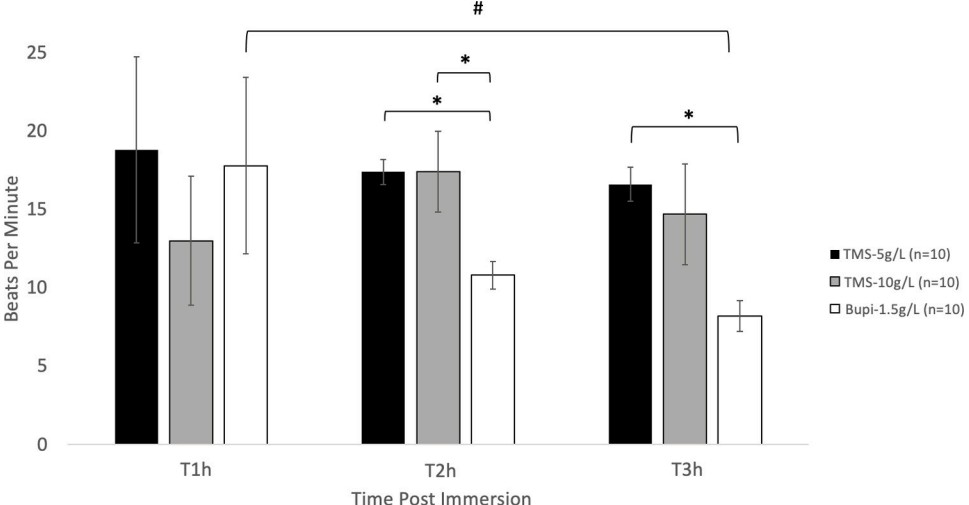

**Fig 2. Heart rate after 1-h immersion.** Heart rate (bpm; mean ± SEM) detected audibly via doppler ultrasound flow evaluation. * values are significantly different ($P < 0.05$) from each other. # values are significantly different ($P < 0.05$) between time points.

at 5 h were returned to their respective solutions and immersed for a total of 19 h. At 19 h, all remaining animals exhibited rigor mortis. Upon gross necropsy all cardiac activity was completely ceased (Fig 3 and Table 1).

## Statistical analysis

To assess significance of differences in heart rates (beats per minute) by group and over time, repeated-measures ANOVA with Bonferroni correction for multiple comparisons (R Development Core Team, 2015) was performed. Data were expressed as mean ± SEM. A *P* value of less than 0.05 was considered significant. To assess significance of differences between groups for number of animals with ceased heart rates after 5-h immersion a Fisher's exact test was performed.

## Discussion

This study demonstrates that: 1) 1-h immersion of African Clawed frogs in Bupi-1.5 has a faster onset to decrease heart rates compared to TMS-5 and TMS-10; 2) neither TMS-5, TMS-10, nor Bupi-1.5 resulted in complete cessation of heart rate after 1-h immersion; 3) at 19-h immersion, complete cessation of heart rate was observed in both TMS-5 and Bupi-1.5.

In the laboratory setting, groups of African Clawed frogs may need to be culled from colonies for health or experimental reasons. The euthanasia method should be reliable and relatively simple to perform. For individual animals, topical or injectable euthanasia methods are

**Table 1. Animals with ceased heart rates after 5- & 19-h immersions.**

|          | T5h  | T19h |
|----------|------|------|
| TMS-5    | 43%  | 100% |
| Bupi-1.5 | 14%  | 100% |

Animals without audible or visual heart beats after 5-h and at 19-h (i.e. overnight) immersion are presented as percentages.

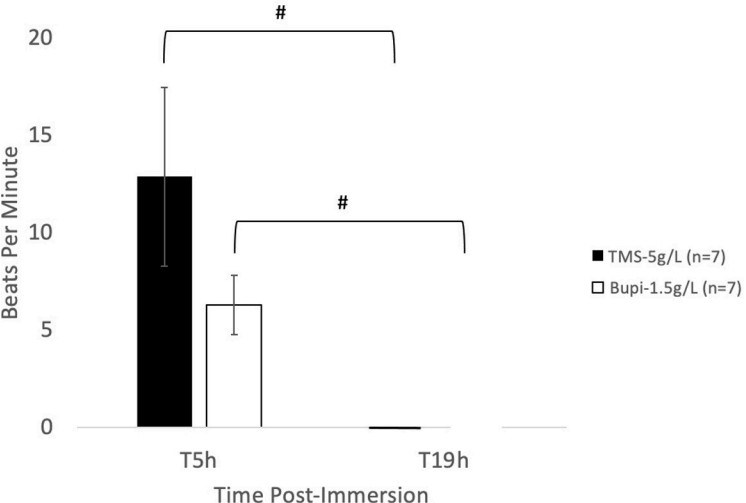

**Fig 3. Heart rate after 5- & 19-h immersions.** Heart rate (bpm; mean ± SEM) detected audibly via doppler ultrasound evaluation. # values are significantly different ($P < 0.05$) between time points.

described, but may be logistically challenging and time consuming if larger groups must be culled [17]. In fish and amphibians, TMS acts centrally on the central nervous system to produce general anesthesia when the body is immersed in solution, and has also been investigated & utilized off-label for euthanasia [5, 6, 18–24]. As discussed in the introduction, there are logistical drawbacks to utilizing TMS for this cause, prompting investigation for an alternative. In mammals, an overdose of bupivacaine is known to be toxic to the cardiovascular system compared to other local anesthetics due to increased lipophilicity, longer duration of blocking sodium channels, and high affinity for cardiac specific sodium channels [25–27]. In dogs, a bupivacaine overdose causes significant hypotension and bradycardia including ventricular arrythmias contributing to cardiovascular collapse and death [28]. We hypothesized we could utilize these effects of bupivacaine overdose for African Clawed frogs. The aim of this study was to determine whether immersion in Bupi-1.5 (i.e. Bupivacaine at 1.5 g/L) would have a similar effect on heart rate as TMS-5 during the euthanasia process in African Clawed frogs.

## 1-hour immersion

For euthanasia of African Clawed frogs it is recommended to immerse animals in TMS-5 for minimum of 1 h, therefore heart rate was assessed immediately after 1 h, then at 2 and 3 h post the 1-h immersion for part one of the experiment. TMS-5, TMS-10, and Bupi-1.5 were utilized. TMS-10 was included as during a pilot study TMS-5 was observed to be insufficient in ceasing heart rate for immersed frogs, thus we expected twice the recommended concentration would be sufficient to achieve this endpoint. The AVMA euthanasia guidelines use irreversibility of death as part of the criteria for evaluating effective euthanasia methods [3]. For our study, we defined death as a total cessation of heart rate (i.e. cardiac death). We utilized an ultrasonic doppler flow detector to audibly evaluate heart rate for all frogs and visually observed heart movement at the end of each experiment and to confirm ceased heart rate.

Surprisingly, the heart rate of TMS-5 and TMS-10 frogs did not differ at any time point (Fig 2). A previous euthanasia study utilizing African Clawed frogs showed complete cessation of cardiac activity for all frogs at 3 h post a 1-h immersion in a TMS-5 solution [5]. However, in this current study, heart rates did not show complete cessation at 3 h post the 1-h immersion in either TMS-5 or TMS-10 groups (heart rates were 16.6 and 14.7 bpm, respectively).

Although the definitive reason for this disparity between this current study and the 2009 study is unclear, the differences in age of the frogs may have contributed to the difference in outcome. Concentration and duration of TMS to effectively euthanize African Clawed frogs has also been shown to vary by life stage [23]. Whereas all frogs utilized in the 2009 study were aged 2–3 years old, we included retired breeders from various colonies whose age was not specified. Expected breeding years for female African Clawed frogs can range from 3–6 years, but frogs have been kept in laboratories for egg production for 12 years or more; it is very likely a number of our frogs exceeded 3 years of age [2].

In this current study, although it did not cease heart rate, Bupi-1.5 treated animals had lower heart rates compared to TMS-5 at 2 and 3 h post 1-h immersion, and lower heart rates compared to TMS-10 at 2 h (Fig 2). Bupivacaine is 2–4 times more cardiovascular toxic than other local anesthetics in mammals (e.g. lidocaine), hence it was selected for this current study, however a cardiovascular toxic dose for African Clawed frogs has not been established [28]. The concentration of bupivacaine in this experiment was selected based on data from a pilot study we conducted in which frogs were immersed in several concentrations of a bupivacaine solution and heart rate was assessed. In this current study the decreased heart rate of Bupi-1.5 animals compared to the TMS-5 and TMS-10 groups at any time point was likely due to overdose of bupivacaine. At 3 h post the 1-h immersion, none of the groups showed complete cessation of heart rate. The lack of cardiac death after the 1-h immersion at any time point in TMS-5, TMS-10, and Bupi-1.5 solutions for this study is strong support for the use of a secondary physical method of euthanasia after 1-h immersion. It should also be noted while bupivacaine does not pose the same potential eye hazard to humans as TMS, multiple vials had to be drawn up to create the 1.5 g/L solution, which should be factored in for time and labor.

## 5- and 19-hour immersion

Because complete cardiac cessation was not observed in the first experiment by 3 h post the 1-h immersion, we extended immersion time in this second experiment. This portion focused on TMS-5 as this dose is recommended within the AVMA Euthanasia guidelines and Bupi-1.5. Frogs were immersed for a longer duration at 5 or 19 h. A 5-h immersion represents a time scheme wherein animal care or laboratory personnel may begin a large group euthanasia at the start of a work day and complete the task before the end of the work day, while the 19-h immersion represents an overnight immersion. Heart rates were assessed utilizing the same method as experiment one immediately at each time point. Although the 5-h immersion showed no statistical difference in heart rate between TMS-5 or Bupi-1.5, a number of the frogs had ceased heart rate (i.e. absent cardiac sounds, no visual cardiac contractions after celiotomy) in both the TMS-5 and Bupi-1.5 groups (43% and 14%, respectively; Fig 3 and Table 1). It is possible that at the 5-h immersion, although heart rate was not completely ceased, the remaining frogs were on the pathway to brain death or irreversible cardiac collapse. In mammals, cardiac arrest without intervention has been shown to result in loss of brain activity within 30 seconds due to the high glucose and oxygen requirements of neural tissue [29]. However, in aquatic animals and reptiles cardiac activity can persist after brain death due to the myocardium utilizing local glycogen stores, as opposed to blood glucose [4, 8]. Anecdotally, this mechanism can even be observed with the heart removed from the body. However, to technically identify brain death, an electroencephalogram (EEG) must be performed, which is not an accessible tool for animal care personnel. Although EEG has been used to assess plane of anesthesia in TMS anesthetized African Clawed frogs, it was not utilized in this study [30].

After the 5 h heart rate assessment, the immersion time was continued until 19 h. At 19-h immersion, heart rate was not detected in any animals in either TMS-5 or Bupi-1.5 groups. All

remaining frogs exhibited rigor mortis and had absent visible cardiac contractions upon celiotomy. As discussed, determining death using heart contractions in amphibians can be difficult due to the ability for persistent heart contractions after apparent euthanasia, however rigor mortis is a confirmatory sign of death [3]. Additionally, while African Clawed frogs live as aquatic amphibians, they are surface breathers [31]. It is likely some of the frogs died from lack of oxygen as they were under deep general anesthesia and could not surface for air [32]. Because the 19-h immersion represents an overnight euthanasia with the endpoint being death we did not assess heart rate between 5 and 19 h. Although the intention for using TMS and bupivacaine during the 19-h immersion was for the drugs to cause cessation of heart rate, if the animals succumb to lack of oxygen during this time instead, a humane death is still achieved. Both TMS-5 and Bupi-1.5 animals rapidly lost consciousness as indicated by absence of movement soon after immersion and loss of righting reflex, and remain anesthetized until they succumbed to death either from complete cessation of heart rate, and thus collapse of cardiac function, or lack of oxygen. Use of EEG, echocardiogram, and/or cardiac output measurements would aid in identifying if brain death occurs before cardiac death using this euthanasia technique. As drug dosage and efficacy has been demonstrated to differ between sexes in a number of species future studies should also include a male cohort to assess impact of sex on time to cease heart rate [33–35].

In summary, this study indicates that 1) after a 1-h immersion, frogs in Bupi-1.5 had a faster initial onset to decreased heart rates compared to TMS-5 or TMS-10, however no solution ceased heart rate of any animal; 2) at 5-h immersion, 47% of TMS-5 & 17% of Bupi-1.5 treated animals had completely ceased heart rates 3) at 19-h immersion, heart rate of all frogs was completely ceased and all animals exhibited rigor mortis. The inconsistency in producing cardiac death throughout this study with any solution is strong support for the use of a secondary physical method of euthanasia when utilizing an immersive euthanasia method in African Clawed frogs.

## Supporting information

**S1 Table. Data used for Fig 2.**
(TIF)

**S2 Table. Data used for Fig 3.**
(TIF)

## Acknowledgments

We would like to thank Andrea Craig and Alberto Gaudiel for their assistance in care for the animals.

## Author Contributions

**Conceptualization:** David Chu, Cholawat Pacharinsak.

**Data curation:** Kaela Navarro, Katechan Jampachaisri, David Chu.

**Formal analysis:** Katechan Jampachaisri, Cholawat Pacharinsak.

**Funding acquisition:** Kaela Navarro, Cholawat Pacharinsak.

**Investigation:** Kaela Navarro, David Chu, Cholawat Pacharinsak.

**Methodology:** Kaela Navarro, Cholawat Pacharinsak.

**Supervision:** Cholawat Pacharinsak.

**Writing – original draft:** Kaela Navarro.

**Writing – review & editing:** Kaela Navarro, David Chu, Cholawat Pacharinsak.

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
