## [Decision Letter · Decision Letter 0]

1 Sep 2022

PONE-D-22-17202Bupivacaine as a euthanasia agent for African Clawed Frogs (Xenopus laevis)PLOS ONE

Dear Dr. Navarro,

Thank you for submitting your manuscript to PLOS ONE. After careful consideration, we feel that it has merit but does not fully meet PLOS ONE’s publication criteria as it currently stands. Therefore, we invite you to submit a revised version of the manuscript that addresses the points raised during the review process.

We look forward to receiving your revised manuscript.

Kind regards,

Rishabh Charan Choudhary

Academic Editor

PLOS ONE

Journal Requirements:

3. As part of your revision, please complete and submit a copy of the Full ARRIVE 2.0 Guidelines checklist, a document that aims to improve experimental reporting and reproducibility of animal studies for purposes of post-publication data analysis and reproducibility: https://arriveguidelines.org/sites/arrive/files/Author%20Checklist%20-%20Full.pdf (PDF). Please include your completed checklist as a Supporting Information file. Note that if your paper is accepted for publication, this checklist will be published as part of your article.

   "We would like to thank Andrea Craig and Alberto Gaudiel for their assistance in care for the animals. The study was supported by the Stanford Department of Comparative Medicine."

 "This work was supported by Stanford's Department of Comparative Medicine. The funders had no role in study design, data collection and analysis, decision to publish, or preparation of the manuscript. The content is solely the responsibility of the authors and does not necessarily represent the official views of the Stanford Department of Comparative Medicine or any other agency of the State of California."

6. We note that you have indicated that data from this study are available upon request. PLOS only allows data to be available upon request if there are legal or ethical restrictions on sharing data publicly. For more information on unacceptable data access restrictions, please see http://journals.plos.org/plosone/s/data-availability#loc-unacceptable-data-access-restrictions. 

7. Please include your tables as part of your main manuscript and remove the individual files. Please note that supplementary tables (should remain/ be uploaded) as separate "supporting information" files

Reviewers' comments:

Reviewer's Responses to Questions

**Comments to the Author**

1. Is the manuscript technically sound, and do the data support the conclusions?

Reviewer #1: Yes

Reviewer #2: Yes

2. Has the statistical analysis been performed appropriately and rigorously? 

Reviewer #1: Yes

Reviewer #2: Yes

3. Have the authors made all data underlying the findings in their manuscript fully available?

Reviewer #1: No

Reviewer #2: No

4. Is the manuscript presented in an intelligible fashion and written in standard English?

Reviewer #1: Yes

Reviewer #2: Yes

5. Review Comments to the Author

Reviewer #1: Dear Author,

This is an interesting. However ,lease elaborate the practical utility and implication of the present and advantage over other existing models.

How these dat can be used to expand knowledge in clinical or basic research files

Reviewer #2: The author is willing to share raw data upon request, however states that restrictions will apply as far as posting to a public repository. It is not included with the manuscript.

Manuscript review PONE-D-22-17202, entitled "Bupivacaine as a euthanasia agent for African Clawed Frogs (Xenopus laevis)".

Overall Comments: This manuscript describes the immersion of African Clawed frogs in bupivacaine or TMS for humane euthanasia. This information is beneficial to the research community and describes a potential novel euthanasia agent for this species, in that bupivacaine successfully lowered the heart rate more quickly than the standard immersion agent and produced irreversible death. This give the community another, safer, agent to use. It also shows the importance of using a secondary method of euthanasia when following the AVMA Guidelines for the use of TMS in this species. The study is described well and conclusion are appropriate.

Abstract, Line 31- 9 hours is stated, this should be 19.

Introduction, Line 47- Please add the word “and” between endocrinology and toxicology.

Materials and Methods, Line 110- Please add a sentence prior to “A celiotomy was performed…” on how it was confirmed a surgical plane of anesthesia was attained.

Results, Line 139- Please add a reference to Figure 2.

Discussion, Line 228-9- “brain death or irreversible” Please complete this sentence.

Figure 1B. There should be an indication that frogs without a heart beat had a celiotomy performed at the 5h timepoint.

6. PLOS authors have the option to publish the peer review history of their article (what does this mean?). If published, this will include your full peer review and any attached files.

Reviewer #1: **Yes: **Amit Agrawal

Reviewer #2: No

---

## [Author Response · Author response to Decision Letter 0]

10 Sep 2022

Response: This has been revised.

Response: Thank you for the comment; this was included in our manuscript and can be located in the revised version in lines 104-108. The additional requirements of anesthesia and euthanasia can be found within the methods section. 

3. As part of your revision, please complete and submit a copy of the Full ARRIVE 2.0 Guidelines checklist, a document that aims to improve experimental reporting and reproducibility of animal studies for purposes of post-publication data analysis and reproducibility: https://arriveguidelines.org/sites/arrive/files/Author%20Checklist%20-%20Full.pdf (PDF). Please include your completed checklist as a Supporting Information file. Note that if your paper is accepted for publication, this checklist will be published as part of your article.

Response: This checklist has been completed and uploaded as a supplemental file. 

Response: This has been clarified in the resubmission process. 

 "We would like to thank Andrea Craig and Alberto Gaudiel for their assistance in care for the animals. The study was supported by the Stanford Department of Comparative Medicine."

 "This work was supported by Stanford's Department of Comparative Medicine. The funders had no role in study design, data collection and analysis, decision to publish, or preparation of the manuscript. The content is solely the responsibility of the authors and does not necessarily represent the official views of the Stanford Department of Comparative Medicine or any other agency of the State of California."

Response: Thank you. The funding statement in the manuscript has been removed. We have addressed the statement in our cover letter. 

6. We note that you have indicated that data from this study are available upon request. PLOS only allows data to be available upon request if there are legal or ethical restrictions on sharing data publicly. For more information on unacceptable data access restrictions, please see http://journals.plos.org/plosone/s/data-availability#loc-unacceptable-data-access-restrictions. 

Response: We have uploaded supplementary files to provide a minimal data set. All other data are included within the manuscript. 

3. Have the authors made all data underlying the findings in their manuscript fully available?

Reviewer #1: No

Reviewer #2: No

Response: We have uploaded supplementary files to provide a minimal data set. All other necessary data are included within the manuscript. 

5. Review Comments to the Author

Reviewer #1: Dear Author, This is interesting. However, please elaborate the practical utility and implication of the present and advantage over other existing models.

How these data can be used to expand knowledge in clinical or basic research files.

Response: Thank you for your comment. We have added additional information elaborating on the importance of Xenopus as an animal research model, why large group culls may be necessary, and the importance of alternative euthanasia agents in the introduction. 

Reviewer #2: The author is willing to share raw data upon request, however states that restrictions will apply as far as posting to a public repository. It is not included with the manuscript.

Response: We have uploaded supplementary files to provide a minimal data set. All other necessary data are included within the manuscript.

Reviewer 2: Abstract, Line 31- 9 hours is stated, this should be 19.

Response: Thank you, this has been revised. 

Reviewer 2: Introduction, Line 47- Please add the word “and” between endocrinology and toxicology.

Response: Thank you, this has been revised.

Reviewer 2: Materials and Methods, Line 110- Please add a sentence prior to “A celiotomy was performed…” on how it was confirmed a surgical plane of anesthesia was attained.

Response: This has been revised to confirm a surgical plane of anesthesia had been attained prior to celiotomy and how it was performed. 

Reviewer 2: Results, Line 139- Please add a reference to Figure 2.

Response: This has been added. 

Reviewer 2: Discussion, Line 228-9- “brain death or irreversible” Please complete this sentence.

Response: Thank you, this has been revised.

Reviewer 2: Figure 1B. There should be an indication that frogs without a heart beat had a celiotomy performed at the 5h timepoint.

Response: The figure has been updated to reflect the celiotomy at 5h and the figure text has been updated to reflect when celiotomies were performed and for which animals.

---

## [Decision Letter · Decision Letter 1]

5 Dec 2022

Bupivacaine as a euthanasia agent for African Clawed Frogs (Xenopus laevis)

PONE-D-22-17202R1

Dear Dr. Navarro,

We’re pleased to inform you that your manuscript has been judged scientifically suitable for publication and will be formally accepted for publication once it meets all outstanding technical requirements.

Kind regards,

Hazel Bautista

Support Staff - Lead

PLOS ONE

Additional Editor Comments (optional):

Reviewers' comments:

Reviewer's Responses to Questions

**Comments to the Author**

1. If the authors have adequately addressed your comments raised in a previous round of review and you feel that this manuscript is now acceptable for publication, you may indicate that here to bypass the “Comments to the Author” section, enter your conflict of interest statement in the “Confidential to Editor” section, and submit your "Accept" recommendation.

Reviewer #1: All comments have been addressed

Reviewer #2: All comments have been addressed

2. Is the manuscript technically sound, and do the data support the conclusions?

Reviewer #1: Yes

Reviewer #2: Yes

3. Has the statistical analysis been performed appropriately and rigorously? 

Reviewer #1: Yes

Reviewer #2: Yes

4. Have the authors made all data underlying the findings in their manuscript fully available?

Reviewer #1: Yes

Reviewer #2: Yes

5. Is the manuscript presented in an intelligible fashion and written in standard English?

Reviewer #1: Yes

Reviewer #2: Yes

6. Review Comments to the Author

Reviewer #1: The authors have made suggested changes

The practical of study has been added

The data sets have been made available

Reviewer #2: (No Response)

7. PLOS authors have the option to publish the peer review history of their article (what does this mean?). If published, this will include your full peer review and any attached files.

Reviewer #1: No

Reviewer #2: **Yes: **Amanda Darbyshire

---

## [Editor Report · Acceptance letter]

12 Dec 2022

PONE-D-22-17202R1 

Bupivacaine as a euthanasia agent for African Clawed Frogs (*Xenopus laevis*) 

Dear Dr. Navarro:

I'm pleased to inform you that your manuscript has been deemed suitable for publication in PLOS ONE. Congratulations! Your manuscript is now with our production department. 

Kind regards, 

on behalf of

Dr. Michael Klymkowsky 

Academic Editor

PLOS ONE